# Predictors of Burnout and Well-Being Among Veterinarians in Slovenia

**DOI:** 10.3390/vetsci12040387

**Published:** 2025-04-20

**Authors:** Ožbalt Podpečan, Valentina Hlebec, Metka Kuhar, Valentina Kubale, Breda Jakovac Strajn

**Affiliations:** 1National Center for Animal Welfare, Veterinary Faculty, University of Ljubljana, Gerbičeva 60, 1000 Ljubljana, Slovenia; 2Academic Unit of Social Informatics and Methodology, Faculty of Social Sciences, University of Ljubljana, Kardeljeva ploščad 5, 1000 Ljubljana, Slovenia; valentina.hlebec@fdv.uni-lj.si; 3Academic Unit of Media Studies, Faculty of Social Sciences, University of Ljubljana, Kardeljeva ploščad 5, 1000 Ljubljana, Slovenia; metka.kuhar@guest.arnes.si; 4Institute of Preclinical Sciences, Veterinary Faculty, University of Ljubljana, Gerbičeva 60, 1000 Ljubljana, Slovenia; valentina.kubale@vf.uni-lj.si; 5Institute of Food Safety, Veterinary Faculty, Feed and Environment, University of Ljubljana, Gerbičeva 60, 1000 Ljubljana, Slovenia; breda.jakovacstrajn@vf.uni-lj.si

**Keywords:** veterinarian burnout, mental health in veterinarians, work–life balance, burnout predictors, ethical dilemmas

## Abstract

Veterinary medicine is often idealized as a beautiful, romantic and noble profession dedicated to improving the well-being of animals and their owners. Veterinarians are expected to always be available, empathetic, knowledgeable and fully committed to their work. However, such professional commitment comes with considerable challenges. Burnout syndrome is particularly common in caring professions such as veterinary medicine. Burnout symptoms associated with ethical and moral dilemmas, financial stress, loneliness and a disturbed work–life balance often affect the mental health of veterinarians. Although awareness of mental health problems in veterinary medicine is increasing worldwide, studies are mainly focused on Australia and Western regions such as North America and Western Europe. Eastern and Southern Europe, including Slovenia, are notably underrepresented in the literature. This study shows that work–life imbalance, ethical conflicts and long working hours in Slovenian veterinarians are important predictors of burnout symptoms. Younger and female veterinarians are disproportionately affected compared to older veterinarians who are better able to cope with stress.

## 1. Introduction

Veterinary medicine is often idealized by laypeople as a beautiful, romantic and noble profession dedicated to improving the well-being of animals and their owners. Veterinarians are expected to always be available, empathetic, knowledgeable and fully committed to their work. However, such professional commitment comes with considerable challenges. Over the past two decades, numerous studies have shown that veterinarians are exposed to a variety of stress factors that have a negative impact on their mental well-being [1,2,3,4,5,6,7]. Over the last two decades, the profession has undergone significant changes, including increased commercialization, increased specialization and increased client expectations. These changes have increased stress factors that negatively affect the mental health of veterinarians [1,2,3,4,5,6,7,8]. Ethical and moral dilemmas, financial burdens, loneliness and a disrupted work–life balance contribute to high levels of burnout. Alarmingly, these stressors contribute to a higher risk of suicidal thoughts, suicide attempts and completed suicides within the profession [5,9,10,11,12]. One of the most pressing problems is burnout, a multidimensional syndrome characterized by emotional exhaustion, depersonalization and a diminished sense of personal fulfillment. Burnout occurs when the demands of the workplace exceed the individual’s ability to cope with them, resulting in chronic stress, emotional withdrawal and reduced professional effectiveness [13,14,15]. Burnout is particularly common in caring professions such as veterinary medicine, where long working hours, the emotional strain of dealing with clients and frequent ethical dilemmas add to the pressures of daily practice. In veterinary medicine, burnout is increasingly recognized as a critical problem, and there is growing evidence of its far-reaching impact on mental health and career sustainability [7,16,17]. According to the Merck studies on veterinarian well-being, 70–75% of veterinarians reported suffering from moderate to very severe symptoms of burnout, with younger professionals and women disproportionately affected. Veterinarians who felt they had less control over their work were more likely to suffer from burnout, which was exacerbated by longer working hours and increased evening and weekend shifts. It has been shown that US veterinarians suffer from burnout at a similar rate to the general population [9,10,18]. Recent studies on veterinarians working in teaching hospitals shed further light on this problem. Adin et al. [19] used the Stanford Professional Fulfillment Index to assess burnout and professional fulfillment among a clinical faculty in a large academic veterinary department. The study found that 61.7% of faculty met the criteria for burnout, with significantly higher scores for burnout and professional fulfillment compared to academic physicians. Faculty members valued their work and their patients, but cited high workload and lack of autonomy as major stressors. Although burnout is a specific syndrome, it is closely related to broader issues of mental health, which is defined by the World Health Organization [20,21] as “a state of well-being in which a person recognizes their own abilities, can cope with the normal stresses of life, can work productively, and is able to contribute to their community”. This definition emphasizes that mental health is not only the absence of illness, but also includes resilience, emotional well-being and effective functioning in different areas of life. However, the stressors of the veterinary profession—long hours, compassion fatigue and financial pressures—often undermine these qualities. For example, veterinarians often have high student debt, and their salaries may not adequately reflect their level of training and responsibility [18]. As noted by Cake et al. [22], resilience remains an under-researched concept in the veterinary research literature. Burnout and related mental health issues are not only a personal problem, but also have profound implications for workplace productivity, human well-being and public health. These challenges are in line with general global trends where mental disorders are among the main causes of absenteeism. In Slovenia, mental and behavioral disorders are responsible for approximately 1.2 million lost calendar days annually, highlighting the societal and economic burden of untreated mental health problems [23]. According to the Eurostat Labor Force Survey [24], 45% of workers in 27 European Union countries reported being exposed to risk factors that have a potentially negative impact on their mental health. The most-frequently mentioned stress factors included time pressure or high workload (19.3% in the EU and Slovenia) and working with demanding clients, patients or students (10.3% in the EU; 10.0% in Slovenia), followed by job insecurity and inadequate communication within organizations. While research on mental health in veterinary medicine has expanded globally, it remains predominantly focused on Australia and Western regions such as North America and Western Europe. Eastern and Southern Europe, including Slovenia, are significantly underrepresented in the literature [25,26,27]. Although veterinarians from Slovenia participated in two European surveys on well-being and stress levels in 2018 and 2023 [7], the number of Slovenian participants accounted for less than 15% of the total national veterinary workforce. Slovenia’s unique cultural and professional context, characterized by dual-income households, small veterinary practices and a particular legal and health framework, requires targeted research to uncover specific stressors and develop tailored interventions. In addition, there is a growing trend in the current veterinary research landscape towards large-scale, multi-country studies. While these studies provide valuable insights into the general trends impacting the profession, they often overlook the unique cultural, economic and legal context that can significantly influence the experiences and challenges of veterinarians in individual countries or regions. Therefore, there remains a need for more comprehensive, context-specific studies, such as this examination of the Slovenian veterinary profession, to better understand the interplay of these factors and develop more effective strategies to support veterinarians’ mental health.

The aim of this study is to investigate the prevalence and predictors of burnout in veterinarians in Slovenia, with a focus on work–life balance, ethical dilemmas and general well-being. By examining these factors, this study aims to contribute to a more nuanced understanding of veterinarians’ mental health and provide actionable insights to promote resilience, reduce burnout and improve job satisfaction in the profession.

The study has two main objectives: (1) to assess the extent of burnout symptoms among Slovenian veterinarians and (2) to investigate the role of various factors such as work–life balance, ethical dilemmas, and other potential predictors of burnout. By achieving these goals, this study not only improves the understanding of factors contributing to burnout, but also lays the foundation for the development of evidence-based interventions.

## 2. Materials and Methods

The present study is a cross-sectional survey. The observed population includes all registered and licensed veterinarians in Slovenia who are actively working as veterinarians in this profession, but also registered veterinarians who have left the profession for various reasons and are no longer working in this profession. In total, there were about 1250 active veterinarians in Slovenia in 2024. This is a population study, as all registered veterinarians were invited to participate in the survey through the Veterinary Chamber of Slovenia (VCS). This included veterinarians working in private practices, clinics and hospitals as well as in industry, veterinarians employed at the University of Ljubljana of Veterinary Faculty and veterinarians employed by the Administration for Food Safety, Veterinary Sector and Plant Protection. The invitation was sent by e-mail in Slovenian on 13 February 2024, and a reminder was sent on 3 March 2024.

The questionnaire included sections on socio-demographic characteristics (e.g., gender, age, household composition, employment status), work-related aspects (e.g., position at work, workload, financial situation) and indicators of well-being. It included validated measures of burnout and psychological symptoms as well as self-assessments of work–life balance, job satisfaction and physical health.

The data were collected between 13 February 2024 and 11 March 2024. The study and the questionnaire were approved by the Ethics Committee for Research at the Faculty of Social Sciences (No. 801–2023–012/TD). A total of 473 veterinarians responded, which corresponds to a response rate of 38%. The VCS provided population characteristics for comparison with the realized sample. The characteristics of the sample are compared with the population data and shown in the Appendix A.

As the realized sample did not show any major deviations apart from a larger proportion of younger veterinarians and a smaller proportion of older veterinarians, the data were not weighted.

### Data Analysis

SPSS (version 29.0.0.0) was used for the statistical analyses in this study. First, descriptive statistics, e.g., frequency distribution, means and standard deviations, were used to summarize the data and describe the distribution of key variables, including burnout symptoms and demographic factors. A multiple regression analysis was then performed to examine the relationships between these variables and the number of burnout symptoms reported by the veterinarians. This multivariate approach allowed us to identify the most important predictors of burnout while controlling for other factors in the model.

The dependent variable in this study is the burnout score, which was calculated as the average number of statements where respondents chose “yes” when reflecting on their experiences in the past month. This measure is based on the Mayo Clinic Physicians Wellbeing Index [15,28,29], which has been used in previous Merck Animal Health studies on veterinary wellbeing [9,10,18,30]. The burnout items listed in Table 1 refer to the main dimensions of psychological distress and job strain, including emotional exhaustion, feelings of hopelessness, emotional hardening and physical and emotional impairment in daily tasks. These items correspond to aspects of burnout, but also capture broader indicators of stress. The response options for each item were binary, i.e., 1 for “yes” and 0 for “no” with items relating to experiences within the past month. This approach provides a useful but simplified measure of burnout symptoms, emphasizing practicality over multidimensional specificity.

The data presented in Table 1 and Table 2 represent specific sections of a broader questionnaire developed to assess the mental well-being, burnout and related predictors in veterinarians.

The independent variables, including demographic factors such as gender, age and living arrangements, as well as work-related characteristics and subjective assessments of job satisfaction and work–life balance, are shown in Table 2.

## 3. Results

### 3.1. The Dependent Variable: Prevalence of Burnout Symptoms Among Veterinarians in Slovenia

The survey revealed that veterinarians experience varying degrees of burnout and mental stress. In the past month, over half (52.8%) reported feeling burnt out, 47.5% expressed emotional distress related to work, and 41.4% felt down, depressed or hopeless. In addition, 44.7% of respondents struggled with an overwhelming workload, while 50.3% were concerned about their emotional well-being. Physical health restrictions affecting daily tasks were reported by 36.5%. Missing information accounted for 16.7%.

Burnout levels were categorized according to the Merck Animal Health Veterinarian Wellbeing Studies [10,30]. Based on the number of symptoms, 45.5% of respondents were categorized as low burnout, 26.4% as moderate burnout and 28.3% as high burnout. The mean number of symptoms reported was 2.9 (SD = 2.2).

### 3.2. Independent Variables: Demographic and Professional Characteristics

The sample consisted of 52.2% female respondents, representing 63.2% of valid responses, while 29.0% were male (35.0% of valid responses). A small proportion (1.5%) did not specify their gender (1.8% of valid responses). Missing data accounted for 17.3%. The average age was 44.4 years (SD = 11.5, range 27–74 years), with missing data at 17.8%.

In terms of household composition, four-person households were the most common (22.2% of all respondents, 27.1% of valid responses), followed by two-person households (19.9% overall, 24.2% valid) and three-person households (17.5% overall, 21.4% valid). Households with five or more members accounted for 12.4% of the total sample (15.2% of valid responses). Missing data accounted for 20.1%.

The majority (68.5% of valid responses) lived with a spouse or partner, while 63.4% of valid responses lived with children. Among those who had children, two children were the most common (21.8% of all respondents, 42.7% of valid responses), followed by one child (18.0% overall, 35.3% valid) and three or more children (10.8% overall, 21.1% valid). Missing responses accounted for 19.0%.

The financial self-assessment revealed that 58.5% of valid responses stated that they were able to live comfortably, while 8.2% stated that they were living very well. In total, 28.5% of valid responses stated that they could just about make ends meet, 4.1% found it difficult and 0.8% found it extremely difficult to cover their living costs. Missing data accounted for 17.5%. The average rating of the valid responses on financial self-assessment was 3.69 on a scale of 1 (“We are struggling extremely to make ends meet”) to 5 (“We live very comfortably”).

The respondents’ childhood environment varied: 31.5% of the valid responses grew up in a small town, 25.1% in a village, 20.5% on a farm or in a rural house, 13.6% in a large city and 9.5% in a suburb. Missing data accounted for 17.3%.

In terms of occupation, 67.7% of valid responses indicated employment without a managerial role, 22.8% held a managerial position and 16.6% were owners or co-owners of a practice. More than half (56.9% of valid responses) worked 41 or more hours per week, with an average total working time of 46.94 h (SD = 14.10). Missing responses accounted for 13.7%.

Among the respondents, 62.2% work in veterinary practice (clinics), including those at the Veterinary Faculty. Missing data accounts for 11.0% of all respondents.

Satisfaction ratings among valid responses showed job satisfaction at 7.17 (SD = 2.04) and family life satisfaction at 7.24 (SD = 2.46) on a scale from 1 (very dissatisfied) to 10 (very satisfied). Ethical conflicts were rated 3.10 (SD = 1.24), and work–life balance difficulties at 3.21 (SD = 1.32), based on a scale from 1 to 5, where higher scores indicate greater difficulties. Missing data accounted for 19.2%.

### 3.3. Predictors of Burnout Score Among Veterinarians in Slovenia: Results from Multiple Regression

A multiple regression analysis was carried out in the study. First, the assumptions were checked: normality of residuals, heteroscedasticity and multicollinearity, and all were met. The overall model was statistically significant (Table 3), explaining about 30% of the variance in symptoms (R2 = 0.325, R2adj = 0.302, Table 4).

Table 3 presents the model fit statistics, while Table 4 shows the results of the ANOVA, confirming the significance of the regression model (F = 13.820, *p* < 0.001).

The regression coefficients are shown in Table 5. Several predictor variables significantly predict the number of burnout symptoms, while some others show a tendency towards significance with *p*-values around 0.100. Given the small sample size, it is worth considering these trends as they may indicate significant relationships. The number of burnout symptoms increases significantly with the total number of working hours (B = 0.016, β = 0.110, *p* = 0.018), the occurrence of ethical conflicts and dilemmas (B = 0.312, β = 0.180, *p* < 0.001) and an unhealthy balance between work and family life (B = 0.48, β = 0.298, *p* < 0.001). Among the three most influential negative predictors, work–family life balance is the most important, followed by ethical conflicts and dilemmas and total number of working hours. Among the predictor variables with statistical significance around 0.100, gender (female) increases the number of burnout symptoms. Predictor variables that significantly reduce the number of burnout symptoms include satisfaction with family life (B = −0.115, β = −0.135, *p* = 0.015) and satisfaction with current work (B = −0.134, β = −0.124, *p* = 0.017), with satisfaction with family life being the strongest predictor variable. Among the predictor variables with statistical significance around 0.100, gender (being female) slightly increases the number of burnout symptoms, age reduces the number of burnout symptoms and working in a veterinary practice also reduces the number of burnout symptoms.

## 4. Discussion

This study provides a comprehensive overview of the current state of burnout and its predictors among veterinarians in Slovenia, a profession that is increasingly recognized as a risk for mental health problems. With around 1250 active veterinarians in Slovenia, this study provides valuable insights into the systemic and personal factors contributing to burnout. In line with international research findings [7,16,17], a significant proportion of veterinarians in this study reported symptoms of burnout, emotional distress and mental fatigue. The prevalence of burnout in our sample was slightly lower than in the 2021 Merck Animal Health Veterinarian Wellbeing Study [18]. These differences may reflect contextual factors specific to the Slovenian veterinary profession or differences in the system and workplace as well as cultural expectations. Burnout is also a non-negligible issue in Slovenia, affecting not only the veterinary profession but also other professional groups. In the study by Pšeničny [31], over 60% of the 1480 respondents showed signs of burnout and 8% experienced severe stages. Although no significant differences were found between gender, age or education level, managers, healthcare professionals and service personnel were more affected. Although this particular study was based on a non-representative sample of workshops, seminars and online surveys, it reflects trends across Europe, where burnout is a growing problem [32]. Compared to the Slovenian general population [33], veterinarians have higher levels of emotional distress, depression and anxiety symptoms, highlighting the well-documented risks associated with demanding care professions. While burnout occurs in many professions, veterinarians may be particularly at risk due to the high workload, ethical conflicts and the dual responsibility of balancing animal welfare with client expectations.

The multiple regression analysis revealed three main factors that contribute to burnout: work–life imbalance, long working hours and ethical conflicts.

An unhealthy work–family life balance emerged as the strongest predictor of burnout symptoms in veterinarians, emphasizing the challenge veterinarians face in maintaining boundaries between their professional and personal responsibilities. Veterinarians, whose work often involves irregular working hours and emotional demands, find it particularly difficult to maintain a healthy balance [10,17]. Due to limited recovery periods and chronic stress levels, they may experience higher levels of burnout, which in turn can affect mental health [34]. This finding is consistent with the existing literature, which identifies work–life balance as one of the key factors for overall well-being and job satisfaction. Veterinarians who have children or other dependents in their household have a greater need to balance demanding professional tasks with family responsibilities [3]. In Slovenia, family life mostly consists of dual earners, so it is likely that work–life balance is an important issue.

The total number of hours worked was found to be a significant predictor of burnout symptoms. This finding is consistent with previous studies [9,10,17,34] which have shown that long working hours can lead to chronic stress, reduced recovery time and physical and mental exhaustion.

Frequent confrontation with ethical conflicts and dilemmas has been identified as another significant predictor of burnout symptoms. Ethical dilemmas are a major cause of stress and burnout in veterinary medicine, a profession that often involves balancing animal welfare with the expectations and resources of animal owners as well as broader societal and legal considerations [35,36]. The results of our study are consistent with the literature highlighting ethical dilemmas as a major stressor in veterinary practice. Veterinarians may be ethically challenged in several ways: in setting boundaries of care, making autonomous decisions, defining personal scope of practice, using appropriate evidence, and communicating concerns about care [35,37]. In addition, the dual responsibility to the animal and the owner can lead to complex ethical conflicts. For example, a veterinarian may feel compelled to recommend euthanasia for an animal with a poor prognosis, even if the owner desires aggressive treatment. Balancing these conflicting interests can cause considerable stress, especially if the veterinarian feels that the chosen course of action is not in the best interest of the animal. These dilemmas are exacerbated when veterinarians feel inadequately prepared or supported in making these difficult decisions, leading to feelings of isolation and helplessness [7].

### 4.1. Protective Factors Against Burnout

While many veterinarians are at significant risk of burnout, satisfaction with family life and satisfaction with work were found to be protective factors that mitigate burnout symptoms. Veterinarians who reported greater satisfaction with their personal lives and greater enjoyment of their work showed fewer symptoms of burnout, underscoring the importance of a supportive work environment and positive social relationships. These findings are supported by previous research emphasizing the importance of external support systems and job satisfaction in mitigating the effects of workplace stress [38,39,40,41].

Among the marginally significant predictors, gender (female) was associated with a higher risk of burnout symptoms. This finding is consistent with broader trends suggesting that women in caring professions often face particular challenges, including increased emotional engagement and societal expectations [16,42]. Age also showed a protective effect: older veterinarians reported fewer symptoms of burnout, likely due to greater professional experience and established coping mechanisms, and probably because younger veterinarians who experience difficulties leave the profession, leaving only those who have developed the resilience to cope well with the demands of the job and their family responsibilities. In addition, working in a veterinary practice can reduce symptoms of burnout, likely due to the structured environment and support systems, and also the opportunity to directly help animals and see the immediate impact of their work can promote a sense of professional fulfillment, counteracting some stress factors.

### 4.2. Generational Differences and Changing Work Expectations

The younger generations of veterinarians, particularly Generation Y, also known as Millennials (born 1977–1997) [43], seem to bring a set of values and expectations to the profession: They only want to work in environments that align with their core socio-political values, even if it means sacrificing salary. The mentality of millennials in the workplace is best described by the words: “Work hard, play harder, but try to only work where you can see yourself play” [44]. While this approach promotes a sense of purpose and alignment with personal values, it can also lead to increased stress if the reality of the workplace does not meet these expectations, potentially making younger vets more susceptible to burnout.

Addressing the complex mental health challenges faced by veterinarians requires comprehensive solutions that consider both systemic and individual factors. Our findings highlight that work–life balance, ethical conflicts and mental health support are key areas where interventions can significantly reduce the risk of burnout.

### 4.3. Enhancing Work–Life Balance

As work–life balance plays an important role in predicting burnout, it is important to develop strategies to help veterinarians achieve a healthier balance between their professional and personal lives. In practice, veterinary practices should consider flexible working arrangements such as part-time work, flexible working hours or the option of remote working. These measures can give veterinarians better control over their working hours, reduce work–life conflict and thus combat one of the main causes of burnout. It is also important to create a supportive work environment that recognizes these challenges. This could include offering resources such as counseling services, stress management workshops or support groups, and encouraging open discussions about work–life balance in the office to create a more understanding work culture.

### 4.4. Managing Ethical Conflicts and Decision Fatigue

In addition to improving the work–life balance, it is equally important to address ethical conflicts. The strong link between ethical dilemmas and burnout identified in this study underscores the urgent need for targeted interventions to help veterinarians overcome these complex challenges. One effective strategy could be the development of comprehensive ethics training programs to make informed, ethical decisions while managing the emotional toll such decisions can take. However, it is also important to recognize the financial realities of veterinary practices. Ethical decisions cannot be made in isolation; they must be balanced with the economic sustainability of the practice. Veterinary clinics must remain financially viable in order to continue to provide high-quality care and offer veterinarians job security. It is also important to create a supportive work environment where ethical issues can be openly discussed and resolved to alleviate the stress associated with these dilemmas. The establishment of ethics counseling services in veterinary medicine could provide assistance in particularly difficult cases. Such services could significantly reduce the burden of decision-making and alleviate the moral pressure that often accompanies ethical conflicts, thus reducing moral anxiety and decision fatigue.

### 4.5. Promoting Mental and Physical Well-Being

Promoting overall well-being through initiatives such as wellness programs, dedicated mental health days, access to physical health resources and structured resilience training can play a critical role in reducing stress and preventing burnout [45,46]. These approaches not only address immediate mental health issues, but also promote a culture that prioritizes long-term resilience and job satisfaction. Our findings are consistent with Cake and co-workers’ [22] recommendation for a balanced approach to veterinary mental health, which emphasizes the integration of resilience-promoting strategies alongside traditional mental health interventions. By strengthening individual coping mechanisms and creating a supportive work environment, these strategies help to sustain the profession, ultimately benefiting both veterinarians and the wider veterinary community.

### 4.6. Addressing Empathy Overload and Financial Stress

In addition, empathy overload and compassion fatigue, as highlighted by NOMV (Not One More Vet) [47], are a major challenge for veterinarians who have to deal with the emotional needs of both animals and their owners. This often leads to emotional exhaustion and an increased susceptibility to burnout. The introduction of structured debriefings and peer support networks could provide the necessary emotional relief. In addition, training in emotion regulation and coping skills, beginning in veterinary school and continuing throughout professional development, could alleviate this pressure. In addition, financial stress—due to student debt and salaries that may not match the level of training and responsibility—further exacerbates burnout. Financial literacy programs could enable veterinarians to better manage economic challenges.

### 4.7. Career Sustainability and Retention Strategies

Our study found that younger veterinarians are at higher risk of burnout, emphasizing the need for mentorship programs that connect them with experienced professionals. Encouraging career mobility, skills diversification and leadership development can open up alternative career paths, reducing alienation and professional stagnation. In addition, integrating personal cognitive and coping skills into veterinary curricula and continuing these programs throughout veterinary careers could improve mental health, engagement and long-term job satisfaction [48].

### 4.8. Limitations and Future Directions

This study provides valuable insights into the factors contributing to burnout among veterinarians in Slovenia, but some limitations should be noted.

One notable limitation of this study is the conceptualization and measurement of burnout. In contrast to established instruments such as the Maslach Burnout Inventory [14], which measures the three core dimensions of burnout—emotional exhaustion, depersonalization and diminished personal accomplishment—the burnout measurement in this study is based on a self-assessment of specific items reflecting psychological strain and stress at work. These items, defined as measures of burnout in the Merck Wellbeing Study [10,30], include questions on emotional hardening, feelings of hopelessness and physical and emotional impairment in daily tasks. While these indicators are consistent with aspects of burnout, they may not fully capture its multidimensional nature. This distinction should be considered when interpreting the results, as the measurement may lead to a general assessment of distress rather than a precise assessment of burnout symptoms as traditionally defined.

However, the relatively high percentage of missing data is a limitation that should be considered. This issue is likely due to the voluntary nature of participation in the survey and the inclusion of sensitive questions about mental health and well-being. Respondents may have chosen not to answer certain questions because they felt uncomfortable, did not have time or did not consider them relevant. In addition, the length and complexity of the questionnaire may have contributed to incomplete responses in certain sections.

Another limitation is that the data are based on self-reports, which can lead to bias as participants may under or overstate their experiences. The inclusion of objective measures, such as clinical assessments or workplace metrics, in future studies could improve the reliability of the results.

The sample also included a slightly higher proportion of older and middle-aged veterinarians, which may limit the generalizability of the results to younger professionals. Future studies should aim to use a more demographically representative sample.

Considering the relatively small population of Slovenia and the relatively small number of people working in veterinary medicine, conducting longitudinal studies would be a practical and meaningful approach to deepening the understanding of burnout dynamics.

The insights gained could inform systemic and individual strategies to support the mental health and career sustainability of veterinarians and ultimately promote resilience and job satisfaction in the profession. Such research could also help evaluate targeted interventions and provide valuable insights into differences between demographic and occupational subgroups. Beyond burnout, these studies could contribute to a more comprehensive understanding of mental health issues in the profession and enable the development of more tailored and effective strategies to promote the overall well-being and career sustainability of veterinarians.

## Figures and Tables

**Table 1 vetsci-12-00387-t001:** Items in the Burnout Score.

Items
Have you felt burned out from your work?
Are you worried that your work is hardening you emotionally?
Have you often been bothered by feeling down, depressed, or hopeless?
Have you fallen asleep while stopped in traffic or driving?
Have you felt that all the things you had to do were piling up so high that you could not overcome them?
Have you been bothered by emotional problems (such as feeling anxious, depressed, or irritable)?
Has your physical health interfered with your ability to do your daily work at home and/or away from home?

**Table 2 vetsci-12-00387-t002:** Independent variables.

Gender	0—Male, 1—Female
Age	Self-reported age
Living with a spouse	0—no, 1—yes
Living with a child	0—no, 1—yes
Making ends meet	1 We can live very well on our current family income. 2 We can live quite well with the current family income. 3 We can just about live on the current family income. 4 It is difficult to live on the current family income. 5 It is very difficult to live on the current family income.
Are you working as manager or owner (director, dean, head of department, professional leader)	0—no, 1—yes
Total number of working hours including regular hours and all overtime hours	Self-reported number
Working in veterinarian practice including the clinics of the veterinary faculty	0—no, 1—yes
How satisfied you are with family life?	The scale ranged from 1 to 10, with 1 representing “very dissatisfied” and 10 representing “very satisfied”.
How happy you are with current work?	The scale ranged from 1 to 10, with 1 representing “very dissatisfied” and 10 representing “very satisfied”.
Subjective evaluation of the impact of this element on job satisfaction: In my work, I encounter ethical conflicts and dilemmas for which I was not adequately prepared.	1 completely disagree 2 disagree 3 nor disagree neither agree 4 agree 5 completely agree
Subjective evaluation of the impact of this element on job satisfaction: My work family life balance is not adequate.	1 completely disagree 2 disagree 3 nor disagree neither agree 4 agree 5 completely agree

**Table 3 vetsci-12-00387-t003:** Summary of model fit statistics for the multiple regression analysis predicting burnout in veterinarians in Slovenia.

Model	R	R Square	Adjusted R Square	Std. Error of the Estimate
1	0.570	0.325	0.302	1.822

**Table 4 vetsci-12-00387-t004:** Analysis of variance (ANOVA) for the multiple regression analysis to predict the burnout score among veterinarians in Slovenia.

Model		Sum of Squares	df	Mean Square	F	Sig.
1	Regression	518.076	12	43.173	13,820	0.000
	Residual	1074.664	344	3.124		
	Total	1592.739	356			

**Table 5 vetsci-12-00387-t005:** Multiple regression analysis of factors predicting burnout score among veterinarians in Slovenia.

	Unstd. Coeff.		Std. Coeff.	t	Sig.
	B	Std. Error	Beta (β)		*p*
Constant	1.546	0.975		1.586	0.114
Gender	0.385	0.205	0.088	1.881	0.061
Age	−0.014	0.010	−0.077	−1.492	0.137
Living with a spouse	0.072	0.277	0.013	0.260	0.795
Living with a child	0.190	0.212	0.044	0.896	0.371
Making ends meet	0.159	0.148	0.053	1.068	0.286
Working as manager or owner	−0.090	0.248	−0.018	−0.363	0.717
Total number of working hours	0.016	0.007	0.110	2.377	**0.018**
Working in veterinarian practice	−0.381	0.215	−0.086	−1.775	0.077
Happiness with family life	−0.115	0.047	−0.135	−2.455	**0.015**
Happiness with current work	−0.134	0.056	−0.124	−2.391	**0.017**
At my work, I encounter ethical conflicts and dilemmas I was not prepared for.	0.312	0.083	0.180	3.754	**<0.001**
My work family life balance is not good	0.481	0.083	0.298	5.766	**<0.001**

Statistically significant results are shown in bold and the alpha value being used is <0.05

## Data Availability

The data presented in this study are available on request from the corresponding author.

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
