# Peer review of "Predictors of Burnout and Well-Being Among Veterinarians in Slovenia"

_vetsci, 2025, doi:10.3390/vetsci12040387_

Round 1
Reviewer 1 Report
Comments and Suggestions for Authors
With interest, I have been reading the manuscript entitled: Predictors of burnout and well-being among veterinarians in Slovenia. The authors show that work-life imbalance, ethical conflicts, and long working hours in Slovenian veterinarians are important predictors of burnout symptoms.
The introduction contains adequate information about the problem to be examined.
Line 138-140: Move the sentence to the conclusion
Material and methods
Line 162-167 This part moves on to the end of this section with the subtitle statistical method. Also, if you did descriptive statistics, write that in this part as well
Line 170, 198, 241 - Do not begin a sentence with a number; modify. There are more sentences in the paper that begin with a number; change them too
Socio-demographic characteristics—This part should be part of the results. Here, you should describe your questionary in a few sentences. (For example, a questionnaire consists of demographic data containing gender, age, and so on, and questions related to Burnout Score, etc.)
Results
Tables that describe the parts 3.1 and 3.2 are required and then reduce the text
Discussion
Line 338-340: It is repeated from materials and methods, delete
Line 377-380 provide reference
Line 390-393 provide reference
Line 418 and 420- explain in parentheses the generation Y and millennial
Conclusions derived from the results obtained like practical implications
Comments on the Quality of English Language
/
Author Response
Response to Reviewer 1 Comments
Comments and Suggestions for Authors
With interest, I have been reading the manuscript entitled: Predictors of burnout and well-being among veterinarians in Slovenia. The authors show that work-life imbalance, ethical conflicts, and long working hours in Slovenian veterinarians are important predictors of burnout symptoms.
Response: We thank the reviewer for their positive opinion on our manuscript.
Point 1: The introduction contains adequate information about the problem to be examined. Line 138-140: Move the sentence to the conclusion.
Response 1: We are thankful to the reviewer for this positive comment and bringing this to our attention. We have moved the sentence from the introduction (Lines 138-140) to the conclusion, as suggested. This revision improves the flow of the introduction and ensures that the key message is emphasized in the concluding section.
Point 2: Line 162-167 This part moves on to the end of this section with the subtitle statistical method. Also, if you did descriptive statistics, write that in this part as well.
Response 2: Subchapter “Data analysis” was added, with detailed description of statistical analyses, including descriptive statistics.
Point 3: Line 162-167 Line 170, 198, 241 - Do not begin a sentence with a number; modify. There are more sentences in the paper that begin with a number; change them too.
Response 3: Response: We have revised all sentences that began with a number, including those at Lines 170, 198, and 241, to ensure they follow proper grammatical conventions. These sentences have been rephrased for clarity and readability.
Point 4: Socio-demographic characteristics—This part should be part of the results. Here, you should describe your questionary in a few sentences. (For example, a questionnaire consists of demographic data containing gender, age, and so on, and questions related to Burnout Score, etc.)
Response 4: We have moved the socio-demographic characteristics to the results section as requested and we added a brief description of the questionnaire into Materials and Methods subchapter (The questionnaire consisted of sections covering socio-demographic characteristics (e.g., gender, age, household composition, employment status), work-related aspects (e.g., job position, workload, financial situation), and indicators of well-being. It included validated measures of burnout and mental health symptoms, alongside self-assessments of work-life balance, job satisfaction, and physical health.).
Point 5: Tables that describe the parts 3.1 and 3.2 are required and then reduce the text.
Response 5: We have added the tables in the supplementary material and shortened the text accordingly.
Point 6: Line 338-340: It is repeated from materials and methods, delete
Response 6: We are thankful to the reviewer for bringing this to our attention. We have removed the repetitive text between Lines 338-340, as it mirrors content already presented in the "Materials and Methods" section. This revision eliminates redundancy and enhances the clarity of the discussion.
Point 7: Line 377-380 provide reference
Response 7: We are thankful to the reviewer for bringing this to our attention. We have added the appropriate reference to support the statements made in Lines 377-380. The citation now provides evidence to strengthen the claims discussed in this section.
Point 8: Line 390-393 provide reference
Response 8: We are thankful to the reviewer for bringing this to our attention. Similarly, we have included a relevant reference for the claims made between Lines 390-393 to ensure that the discussion is well-supported by the literature.
Point 9: Line 418 and 420- explain in parentheses the generation Y and millennial.
Response 9: We are thankful to the reviewer for bringing this to our attention. We have added explanation in parentheses for "Generation Y" and "Millennial" in Lines 418 and 420, as requested.
Point 10: Conclusions derived from the results obtained like practical implications
Response 10: We have revised the Practical Implications section to provide clearer, actionable recommendations based on our findings. Specifically, we expanded strategies for work-life balance, ethical conflict resolution, and mental health support, introduced financial literacy programs, and emphasized mentorship, career sustainability, and resilience training. These additions ensure that the section directly connects our results to practical applications for veterinary professionals and organizations.
We have uploaded the corrected version the the Reviewer 1 to show all the changes, but the revised version will be sent to the editors (we cannot upload both versions).

Reviewer 2 Report
Comments and Suggestions for Authors
The manuscript with the topic "Predictors of burnout and well-being among veterinarians in Slovenia" (Manuscript ID vetsci-3536103) is written in a professional and scientific language. A topic with huge value for the veterinarians in Eastern and Central Europe.
I would like to suggest some points to be improved:
- On line 164 fix the grammar of the new sentence.
- In MATERIALS AND METHODS
- The text in section “Socio-Demographic Characteristics” is better to be presented in a table- easier for the random reader;
- The sections “Socio-Demographic Characteristics” and “Dependent Variable: Burnout Score” are good to be numerated as 2.1 and 2.2 as they are in RESULTS.
- In RESULTS
- The text in section “1 The dependent variable: Prevalence of burnout symptoms among veterinarians in Slovenia” and “3.2 Independent Variables: Demographic and Professional Characteristics” is better to be presented in a tables;
- On line 260 is good to think haw to change the beginning of the sentence: “Of those who gave valid responses…”.
- Between lines 288 and 292 is better to add or just to implement in the text that information which is related to Table 4 and to reject or to add at the end of the paragraph the text from line 303.
- REFERENCES
50% of the references are for a period of the last 5 years. It would be good to add some more.
Author Response
Response to Reviewer 2 Comments
Comments and Suggestions for Authors
The manuscript with the topic "Predictors of burnout and well-being among veterinarians in Slovenia" (Manuscript ID vetsci-3536103) is written in a professional and scientific language. A topic with huge value for the veterinarians in Eastern and Central Europe. I would like to suggest some points to be improved.
Response: We thank the reviewer for their positive opinion on our manuscript.
Point 1: On line 164 fix the grammar of the new sentence.
Response 1: We are thankful to the reviewer for bringing this to our attention. We have changed the text accordingly.
Point 2: The text in section “Socio-Demographic Characteristics” is better to be presented in a table- easier for the random reader.
Response 2: We have added the tables in the supplementary material and shortened the text
Point 3: The sections “Socio-Demographic Characteristics” and “Dependent Variable: Burnout Score” are good to be numerated as 2.1 and 2.2 as they are in RESULTS.
Response 3: As we have presented the socio-demographic characteristics in a table as suggested and shortened the text, the numbering is no longer relevant.
Point 4: The text in section “1 The dependent variable: Prevalence of burnout symptoms among veterinarians in Slovenia” and “3.2 Independent Variables: Demographic and Professional Characteristics” is better to be presented in a tables.
Response 4: We have added these tables into the supplementary material.
Point 5: On line 260 is good to think haw to change the beginning of the sentence: “Of those who gave valid responses…”.
Response 5: Thank you for your suggestion. We have revised the sentence beginning on line 260 to improve its structure.
Point 6: Between lines 288 and 292 is better to add or just to implement in the text that information which is related to Table 4 and to reject or to add at the end of the paragraph the text from line 303.
Response 6: We have integrated the information from Table 4 into the text between lines 288 and 292, ensuring a logical flow. Additionally, we moved the statement about model significance into the paragraph to align with the reviewer's suggestion. Let us know if further adjustments are needed.
Point 7: 50% of the references are for a period of the last 5 years. It would be good to add some more.
Response 7: 4 references for the period of the last 5 years were added. See References: 36, 37, 45, 46.
We have uploaded the corrected version the the Reviewer 1 to show all the changes, but the revised version will be sent to the editors (we cannot upload both versions).

Reviewer 3 Report
Comments and Suggestions for Authors
This is an interesting study and it contributes to the wider discourse regarding veterinary practitioner well being, mental health issues and burnout. It covers an unstudied population and the findings will add to the grow evidence about the mental health risks to veterinary practitioners.
There are some areas that can be improved.
The presentation of the results is confusing and there are some missing methodological details. Of the most significance is a lack of explanation for how the survey data were handled. It is not clear on what basis individual surveys were assessed as being suitable for inclusion/analysis and the criterion for inclusion/exclusion of answers. needs to be explained. It appears that no survey answer was discarded in its entirely, but rather, some questions were not answered by some respondents. These data are then presented as percentages, rather than as actual numbers of missing responses.
Given the small sample size to begin with, this approach obscures the actual numbers involved in the answering of each question leaving it to the reader to undertake manual calculations to gain an accurate insight into how many valid responses per question there were. This is critical information as the authors note in their limitations section. Consequently, these data should be provided, either in a table or as a supplementary materials so readers can clear see how many completed answers were provided for each question. The full survey should also be provided- as administered with the original question order, in the supplementary materials or at the conclusion of the paper. If it represents a translation, please indicate this.
The discussion includes a number of instances where the results are simply recapitulated without analysis or interpretation and these instances should be either deleted or an analysis provided. There is also far too much recapitulation of the numerical results -this should be minimised unless a very specific point is being made about the data.
The inclusion of some graphs or tables to summarise the key results would also aid the reader to understand what is being reported.
There are a number of places were unsubstantiated statements are made that require citations to be added.
I have also made a small number of suggestions in regards to improving the clarity of the writing.

Author Response
Response to Reviewer 3 Comments
Comments and Suggestions for Authors
This is an interesting study and it contributes to the wider discourse regarding veterinary practitioner well being, mental health issues and burnout. It covers an unstudied population and the findings will add to the grow evidence about the mental health risks to veterinary practitioners. There are some areas that can be improved.
Response: We thank the reviewer for their positive opinion on our manuscript.
Point 1: The presentation of the results is confusing and there are some missing methodological details. Of the most significance is a lack of explanation for how the survey data were handled. It is not clear on what basis individual surveys were assessed as being suitable for inclusion/analysis and the criterion for inclusion/exclusion of answers. needs to be explained. It appears that no survey answer was discarded in its entirely, but rather, some questions were not answered by some respondents. These data are then presented as percentages, rather than as actual numbers of missing responses.
Response 1: We have added tables in the supplementary material that clearly present the number of missing responses for each variable, ensuring transparency in data handling. No surveys were entirely excluded; instead, missing values were accounted for in the analysis and are now explicitly reported in the supplementary tables. Let us know if further clarification is needed.
Point 2: Given the small sample size to begin with, this approach obscures the actual numbers involved in the answering of each question leaving it to the reader to undertake manual calculations to gain an accurate insight into how many valid responses per question there were. This is critical information as the authors note in their limitations section. Consequently, these data should be provided, either in a table or as a supplementary materials so readers can clear see how many completed answers were provided for each question. The full survey should also be provided- as administered with the original question order, in the supplementary materials or at the conclusion of the paper. If it represents a translation, please indicate this.
Response 2: The response rate was 38% as described in chapter 2 Materials and methods and realized sample 473, which is not such small number, given that the total population is only 1250. The description of relevant variables of this manuscript are presented Table1 and Table 2. The presentation includes question wordings as well as response categories for each individual variable. Additional presentation of the items in form of survey questionnaire would be just a copy of the information already presented. Therefore, we will only keep the presentation which is already a part of the manuscript. The descriptive statistics is now included as supplement in tables, presenting the absolute and relative frequencies as well as missing values.
Point 3: The discussion includes a number of instances where the results are simply recapitulated without analysis or interpretation and these instances should be either deleted or an analysis provided. There is also far too much recapitulation of the numerical results -this should be minimized unless a very specific point is being made about the data.
Response 3: We have revised the Discussion section to minimize numerical recapitulation and strengthen analysis and interpretation. Instead of simply restating results, we now focus on key predictors of burnout, protective factors, and broader implications. The section has been restructured to enhance clarity, ensuring that findings are critically examined rather than just repeated. Additionally, we have refined sentence structures for better readability and coherence. Let us know if further adjustments are needed.
Point 4: The inclusion of some graphs or tables to summarise the key results would also aid the reader to understand what is being reported.
Response 4: We appreciate the suggestion to include additional graphs or tables. However, given that key numerical findings are already presented in the Results section and detailed in supplementary tables, we believe adding further visualizations within the Discussion may lead to redundancy. Instead, we have focused on providing a clearer synthesis and interpretation of the results. If needed, we can consider including a summary table or figure in the Results section to enhance readability. Let us know if you would like us to proceed in this direction.
Point 5: There are a number of places were unsubstantiated statements are made that require citations to be added
Response 5: Thank you for pointing this out. We have carefully reviewed the manuscript and identified the places where unsubstantiated statements were made. We have added the appropriate citations to support these claims and ensure that all statements are well-supported by the existing literature. We appreciate your attention to this detail, which has strengthened the credibility of our manuscript.
Point 6: I have also made a small number of suggestions in regards to improving the clarity of the writing.
Response 6: We thank reviewer for helpful suggestions regarding the clarity of the writing. We have carefully reviewed the points you raised and made the necessary revisions to improve the manuscript's clarity.
Below, we specifically address the three questions raised in the PDF version:
How were responses determined to be valid? Please elaborate and detail the total no of valid responses compared to all responses received. Then only need to report the percentages of valid responses.
Because of the way these data are being reported it is imperative that the actual numbers are included, not just the percentages.
This needs to be presented as the number of surveys that were discarded due to incomplete answers.
Response: We have clarified how responses were determined to be valid and provided the total number of valid responses compared to all responses received. To ensure transparency, we have added tables in the supplement that explicitly present the number of missing responses for each variable, as well as the number of surveys discarded due to incomplete answers. Additionally, we have revised the results section to report percentages based on valid responses while retaining actual numbers where necessary. Let us know if any further adjustments are needed.
We have uploaded the corrected version the the Reviewer 1 to show all the changes, but the revised version will be sent to the editors (we cannot upload both versions).
